# Expression and Characterization of Two α-l-Arabinofuranosidases from *Talaromyces amestolkiae*: Role of These Enzymes in Biomass Valorization

**DOI:** 10.3390/ijms241511997

**Published:** 2023-07-26

**Authors:** Juan A. Méndez-Líter, Laura I. de Eugenio, Manuel Nieto-Domínguez, Alicia Prieto, María Jesús Martínez

**Affiliations:** 1Department of Microbial & Plant Biotechnology, Centro de Investigaciones Biológicas Margarita Salas, Spanish National Research Council (CSIC), C/Ramiro de Maeztu 9, 28040 Madrid, Spain; jh_mendezliter@hotmail.com (J.A.M.-L.); lidem@cib.csic.es (L.I.d.E.);; 2The Novo Nordisk Foundation Center for Biosustainability, Technical University of Denmark, 2800 Kongens Lyngby, Denmark; manudo@biosustain.dtu.dk

**Keywords:** glycoside hydrolases, hemicellulose, GH62, biodegradation

## Abstract

α-l-arabinofuranosidases are glycosyl hydrolases that catalyze the break between α-l-arabinofuranosyl substituents or between α-l-arabinofuranosides and xylose from xylan or xylooligosaccharide backbones. While they belong to several glycosyl hydrolase (GH) families, there are only 24 characterized GH62 arabinofuranosidases, making them a small and underrepresented group, with many of their features remaining unknown. Aside from their applications in the food industry, arabinofuranosidases can also aid in the processing of complex lignocellulosic materials, where cellulose, hemicelluloses, and lignin are closely linked. These materials can be fully converted into sugar monomers to produce secondary products like second-generation bioethanol. Alternatively, they can be partially hydrolyzed to release xylooligosaccharides, which have prebiotic properties. While endoxylanases and β-xylosidases are also necessary to fully break down the xylose backbone from xylan, these enzymes are limited when it comes to branched polysaccharides. In this article, two new GH62 α-l-arabinofuranosidases from *Talaromyces amestolkiae* (named ARA1 and ARA-2) have been heterologously expressed and characterized. ARA-1 is more sensitive to changes in pH and temperature, whereas ARA-2 is a robust enzyme with wide pH and temperature tolerance. Both enzymes preferentially act on arabinoxylan over arabinan, although ARA-1 has twice the catalytic efficiency of ARA-2 on this substrate. The production of xylooligosaccharides from arabinoxylan catalyzed by a *T. amestolkiae* endoxylanase was significantly increased upon pretreatment of the polysaccharide with ARA-1 or ARA-2, with the highest synergism values reported to date. Finally, both enzymes (ARA-1 or ARA-2 and endoxylanase) were successfully applied to enhance saccharification by combining them with a β-xylosidase already characterized from the same fungus.

## 1. Introduction

Lignocellulose, present in plant biomass, is the most abundant carbon source in nature and the main renewable resource in the biosphere. It is the essential component of the secondary cell walls of plants and is defined as a dynamic and complex structure of cellulose microfibers embedded in a hard hemicellulose matrix reinforced by the presence of lignin [1]. The balance between these three components varies depending not only on the plant species but also on the tissue selected and the stage of development of the organism [2]. Hemicelluloses are a poorly exploited component of lignocellulosic biomass, probably due to its heterogeneous nature, thus needing a variety of different enzymes for its complete degradation [3]. The term hemicellulose includes a series of polysaccharides with heterogeneous structures, generally classified into four types: xylans, mannans, xyloglucans, and mixed-linked β-glucans, which differ from each other in their main structure composition, their branches, and the type and distribution of glycosidic linkages [4].

Xylans are the most common hemicellulose and the second most abundant plant polysaccharide after cellulose [5]. Xylans are branched heteropolysaccharides formed by a main chain of xylose molecules linked by β-1,4 glycosidic bonds and different side chains [5]. If the side chains have glucuronic acid residues, the polysaccharide will be called glucuronoxylan. If it also contains arabinose, it is called glucuronoarabinoxylan, or arabinoxylan, if the proportions of arabinose are higher. A particularity of arabinoxylans is that they may have phenolic units, mainly ferulic acid, linked to carbon 5 of the arabinoses through an ester bond. As a result, a mixture of glycosyl hydrolases and esterases, including endoxylanases, β-xylosidases, α-l-arabinofuranosidases, α-D-glucuronidases, acetyl xylan esterases, and feruloyl esterases, are required for the biodegradation of various forms of xylan. α-l-arabinofuranosidases are glycosidases that hydrolyze the bond between an α-l-arabinofuranosyl substituent and a xylose unit of the xylan or xylooligosaccharide backbone, thus enhancing the action of other enzymes, such as β-D-xylosidase and endoxylanases [6]. These enzymes are exo-type proteins and act from the non-reducing end of the arabinofuranose branches, being able to hydrolyze α-1,2, α-1,3, and α-1,5 bonds. α-l-arabinofuranosidase activity can be found in families GH2, GH3, GH43, GH51, GH54, and GH62. Interestingly, GH62 is the only GH family that contains only arabinofuranosidases, with 24 characterized members, according to the Carbohydrate-Active Enzymes database (CAZy) [7]. The biochemical characterization of GH62 enzymes has led to their definition as arabinoxylan-L-arabinofuranohydrolases since they specifically cut the α-1,2- or α-1,3-l-arabinofuranosidic bonds directly on the arabinoxylan polysaccharide, instead of having preference for arabinoxylooligosaccharides, as occurs in the α-l-arabinofuranosidases of the other families.

The ascomycete fungus *Talaromyces amestolkiae* was postulated as an excellent option not only for cellulose degradation [8] but also for depolymerization of hemicelluloses. In this context, native and recombinant endoxylanases (from families GH10 and GH11), as well as a GH3 β-xylosidase from this fungus, have been characterized and used in different biotechnological applications [9,10,11,12]. However, the presence of accessory enzymes involved in xylan degradation, like α-l-arabinofuranosidases, was not explored to date, although its presence was detected in *T. amestolkiae* secretomes [8]. In this work, two GH62 α-l-arabinofuranosidases (ARA-1 and ARA-2) identified in the genome of this fungus were successfully produced in *Pichia pastoris*. The purification and in-depth characterization of the recombinant enzymes have been carried out. In addition, they have been assayed for enhancing xylooligosaccharides release from arabinoxylan, in combination with a *T. amestolkiae* endoxylanase, as well as for hemicellulose saccharification by using the same enzyme combination and an additional β-xylosidase from the same fungus.

## 2. Results and Discussion

### 2.1. Cloning and Production of ARA-1 and ARA-2

In a previous work, the arabinosidases XM_040874772.1 and XM_040875155.1 were detected in *T. amestolkiae* secretomes obtained from cultures of the fungus in different carbon sources, and they were annotated in the fungal genome [8]. The genes of the two putative GH62 enzymes were renamed *ara-1* and *ara-2*. The gene *ara-1* comprises 1008 bp, with a signal peptide region coding for 57 bp (Appendix A). The mature sequence of the ARA-1 enzyme contains 316 amino acids, has an expected molecular mass of 33.7 kDa, and has a catalytic domain with high identity to other GH62 enzymes. The gene *ara-2* is 1223 bp long with a 54 bp signal peptide and one intron. The mature sequence of the ARA-2 enzyme is 368 amino acids long, resulting in a 39 kDa protein that contains a GH62 catalytic domain and a family 1 N-terminal carbohydrate-binding module (CBM) connected by a Ser/Thr-rich linker (Appendix A). 

The *ara-1* and *ara-2* genes were introduced in the *Pichia pastoris* GS115 strain without their signal peptides to express ARA-1 and ARA-2 following the protocol used for other GHs from *T. amestolkiae* [9]. After the initial screening, clones with the highest activity levels were selected for production, purification, and characterization of the recombinant proteins. ARA-1 activity was maximum (42 U/L) on day 8, while the highest ARA-2 activity (80 U/L) was obtained on day 9 (Figure 1).

### 2.2. Purification of ARA-1 and ARA-2

Enzymatic crudes containing ARA-1 or ARA-2, obtained when *P. pastoris* culture supernatants showed maximal activity, were concentrated and loaded into an anion exchange cartridge (Appendix A). In both cases, the enzyme activity was found in the non-retained fractions before applying the salt gradient. The homogeneity of the enzymes was confirmed by SDS-PAGE (Figure 2). Bands of approximately 37 kDa and 50 kDa were detected for ARA-1 and ARA-2, respectively. The difference detected with their theoretical molecular masses could be due to O- and N-glycosylation, which is common in proteins heterologously expressed in *P. pastoris* [13]. In fact, when both proteins were deglycosylated, their molecular mass decreased to around 30 and 45 kDa, indicating that ARA-1 and ARA-2 are glycosylated proteins with approximately 19 and 10% N-glycosylation, respectively. The higher glycosylation degree of ARA-1 was confirmed by MALDI-TOF since the typical profile of glycosylated proteins observed for both enzymes was much more pronounced in the spectrum of ARA-1 (Figure 2). In both cases, the purification yield was approximately 70% (Table 1).

### 2.3. Physicochemical Properties of ARA-1 and ARA-2

ARA-2 optimal temperature was 60 °C, higher than that of ARA-1 (50 °C). Also, ARA-2 showed higher tolerance to temperatures, retaining approximately 50% activity after 72 h at 50 °C (Figure 3). Regarding the pH profile, ARA-1 was stable in a range from 3 to 7 during 72 h, while its optimum pH was 4, decreasing sharply above it, with non-detectable activity at pH 6. The optimum pH for ARA-2 was also 4, while it remained stable over the entire range measured, confirming the robustness of this enzyme, especially when compared with ARA-1 (Figure 4).

It is important to consider that the optimum temperature and pH for enzymatic saccharification of these substrates require high stability of the enzymes. The properties observed in the α-l-arabinofuranosidases of various fungal species are quite similar to those exhibited by ARA-1 and ARA-2 in this study. However, ARA-1 showed very poor stability within the range of temperatures that were tested. This is a disadvantageous characteristic, as many industrial processes catalyzed by xylanolytic enzymes are typically conducted at high temperatures [14]. On the other hand, the pH stability profiles of ARA-1 and ARA-2 were similar to those of most GH62 fungal α-l-arabinofuranosidases, remaining active and stable at acidic pH levels [15].

As can be observed in Table 2, most GH62 enzymes have optimum pH between 4 and 6.5 and optimum temperature around 40 °C. Although the thermal stability values of ARA-1 and ARA-2 are comparable to those described for other proteins of this family, it is worth noting that the stability of the *T. amestolkiae* enzymes was assessed after 72 h of incubation, while for those reported in the literature, treatment time was generally around 45–60 min. This suggests that ARA-1 and ARA-2 are more resistant, and ARA-2 stands out for its robustness thanks to its ability to withstand pH variations and its high thermostability.

Regarding the effect of different chemical compounds on ARA-1, most of the compounds had an inhibitory effect on the enzyme since activity barely reached 50% in the presence of many of them (Appendix A). Fe_2_(SO_4_)_3_, HgCl_2_, CuSO_4_, AgNO_3_, Pb(NO_3_)_2_, ZnSO_4_ dramatically reduced the activity of ARA-2, while CoCl_2_, MnSO_4_, and FeSO_4_ produced mild inhibition. The absence of inhibition in the presence of EDTA suggests that the protein does not require metal cations for its activity. DTT and 2-ME had no appreciable effect on enzyme activity, so this protein probably does not have disulfide bridges near the active center.

The addition of certain metal ions, such as Ni^2+^, Zn^2+^, Cu^2+^, and Hg^2+^, was found to significantly inhibit the activity of the α-l-arabinofuranosidases. This is a common characteristic of heavy metals, as they tend to bind to free -SH groups found on cysteine residues [15,23]. The sensitivity of the enzymes to Hg^2+^ suggests the presence of a cysteine residue near the active site, although it has also been reported to react with histidine and tryptophan residues. Interestingly, ARA-1 maintains high activity in the presence of Ca^2+^, and strong inhibition occurs upon the addition of EDTA, a chelating agent that sequesters calcium. This suggests that this protein uses calcium as a cofactor, so this divalent cation could play a role in its catalytic activity. While coordination with calcium has been described for other enzymes in the GH62 family, it was previously thought to have a more structural than functional role [24].

### 2.4. Substrate Specificity and Kinetics of ARA-1 and ARA-2

The kinetic parameters determined for ARA-1 against several substrates (Table 3) showed a lower *K_m_* value (greater affinity) and higher *k_cat_* value toward arabinoxylan. In addition, the catalytic parameters indicated that, in terms of affinity and maximum activity, arabinan is a better substrate than *p*NP-AF.

The values calculated for ARA-2 were similar since the enzyme showed a much higher affinity for arabinoxylan in comparison with *p*NP-AF and arabinan. This was also observed for other arabinofuranosidases as that from *A. nidulans* [17], while others are more active toward the *p*NP derivative than the one from *Phanerochaete chrysosporium* [25]. ARA-1 and ARA-2 affinity toward arabinoxylan and arabinan was much better than that from *A. nidulans* arabinofuranosidase, although their *k_cat_* was lower [17].

The GH62 family is unique since it exclusively groups arabinofuranosidases, and according to the CAZy database, only 24 members of this family have been characterized to date. Typically, the specificity of these enzymes is evaluated using three different substrates: *p*NP-AF, wheat arabinoxylan, and arabinan. The catalytic efficiency of ARA-1 on wheat arabinoxylan is one of the highest reported, only surpassed by ABF62b from *P. funiculosum* [20] (Table 2). While all characterized GH62 enzymes are known to degrade arabinoxylans from various sources, this is not always observed for arabinan, whose degradation is harder for these enzymes. The case of *p*NP-AF is particularly interesting because it is considered a model substrate for the rapid detection of arabinofuranosidase activity in all families. However, GH62 enzymes often have difficulties degrading this substrate compared to arabinofuranosidases from families GH43, 51, or 54. The catalytic efficiencies for the degradation of *p*NP-AF range from 0.005 to 1 s^−1^ mM^−1^, but recent research by Linares et al. [21] identified four highly efficient GH62 AFs from *P. subrubescens* that can degrade *p*NP-AF faster, offering new possibilities for discovering novel, robust, and powerful catalysts in the GH62 family. 

Nevertheless, it is important to note that both ARA-1 and ARA-2 showed activity on arabinan, arabinoxylan, and *p*NP-AF, although with a notable difference in activity against the hemicellulose polysaccharides (Table 2). This can be explained by the ability of GH62 α-l-arabinofuranosidases to interact with xylopyranose chains of various lengths through several residues. This interaction helps to orient the catalytic residues on arabinofuranose, making them more active on arabinoxylan than on other substrates with which they cannot interact. Arabinan, on the other hand, has a different structure from xylan because it presents a main chain of arabinofuranoses linked by α (1→5) bonds. This difference in structure probably hinders the interaction between the region of the polysaccharide’s main chain and the binding pocket that all GH62 enzymes possess [26]. 

Finally, it is important to note that functional comparisons of substrate specificity are quite challenging due to the limited number of characterized GH62 enzymes in the literature, and identifying the most efficient enzymes is not straightforward. Although comprehensible due to the lack of substrates and the insoluble nature of some of them, it is essential to expand the pool of GH62 arabinofuranosidases to discover the unique characteristics of these enzymes and their efficiency in degrading lignocellulosic residues.

### 2.5. Cellulose-Binding Assay

As described in the literature, family 1 CBMs can bind to cellulose and, in some cases, to chitin [27]. Considering the presence of one of these modules in ARA-2, the ability to bind cellulose of ARA-1 and ARA-2 was examined and compared. After 2 h incubation, 93% of ARA-1 activity was found in the supernatant, while only 38% ARA-2 activity was in the soluble fraction (Figure 5), showing that the ARA-2 CBM was functional and may have a role in lignocellulose degradation.

In a previous study that extensively analyzed the GH62 family, it was found that 72% of the GH62 genes deposited in the CAZy database possess a carbohydrate-binding module (CBM) [24]. Although these motifs are non-catalytic domains, they play a crucial role in facilitating contact between catalytic domains and substrates, which can increase the hydrolysis rate. Additionally, some CBMs are known to destabilize the crystallinity of polysaccharide chains, thereby enabling enzyme-substrate accessibility [28]. As reviewed in a previous study [24], 52% of GH62 members have one or two CBM13, but only a few GH62 members are linked to CBM1 (8%). To the best of our knowledge, this is the first time that a GH62 enzyme possessing a CBM1 module has been characterized. However, further research will be required to explore the particularities related to this phenomenon.

### 2.6. Synergistic Effect of Arabinofuranosidases with Endoxylanase in the Production of XOS

Due to the heterogeneous nature of hemicelluloses from lignocellulosic biomass, a synergistic effect of hemicellulolytic enzymes has been reported [29]. The effect of arabinofuranose (AF) side chains on xylooligosaccharides (XOS) liberation from arabinoxylan was studied using the GH11 endoxylanase XynM from *T. amestolkiae* recombinantly produced in *P. pastoris*. The activity of GH11 endoxylanases is limited to unsubstituted regions of xylan [30], and then, the release of arabinose substitutions could significantly improve linear XOS production (preferred as prebiotics [31]) from a highly substituted substrate such as arabinoxylan (arabinose:xylose = 38:62).

To analyze its role in XOS production, 0.7% arabinoxylan was preincubated with ARA-1 and ARA-2, and then XynM was added to the reaction. The amount of XOS, determined by HPLC, is shown in Figure 6. Both ARA-1 and ARA-2 significantly increased XynM-catalyzed release of XOS. Higher levels were obtained when incubated with ARA-1, and xylotriose and xylotetraose were found as the most abundant xylooligosaccharides. The yield of total XOS release from arabinoxylan was 9.2 and 6.4% when ARA-1 and ARA-2 were added, respectively. In both samples, xylose levels were found to be below 0.2%. Alvarez et al. [32] reported similar XOS yields, although they also added feruloyl esterases to improve biomass hydrolysis.

On the other hand, the important synergistic effect observed when ARA-1 or ARA-2 were combined with XynM is noticeable, as well as the ratio between the activity of the enzymes together versus the sum of each one used individually. Considering the values of X4 and X5 (no xylotriose was detected when arabinoxylan was incubated with XynM alone), the synergy degree found was 20.2 for ARA-1 and XynM and 13.9 for ARA-2 and XynM. These are much higher values than those reported in the literature [33,34].

### 2.7. Role of ARA-1 and ARA-2 in Saccharification Processes

Besides its role in XOS production, arabinofuranosidases have been described as helper enzymes for bioethanol-oriented saccharification of lignocellulose. The release of the arabinose that decorates the xylose chain could improve the sugars yield from enzymatic hydrolysis, which is necessary to take advantage of biomass [35,36,37]. Also, arabinose could be metabolized by new strains able to ferment pentoses, which increases the productivity of the process [38,39]. In order to maximize xylose release from arabinoxylan, the β-xylosidase BxTw1 from *T. amestolkiae* (0.86 μg/mL) was added to the XOS production reactions (Figure 7). The addition of BxTw1 increased xylose yield from arabinoxylan from 0.2 to 7.9% for ARA-1 and from 0.1 to 11.3 for ARA-2, although detectable amounts of X3–X5 remained in the samples. To enhance the total saccharification of arabinoxylan, an additional pulse of the β-xylosidase BxTw1 was added to the reactions (0.43 μg/mL), which were further incubated for 96 h. The results were remarkable, since the final yields of xylose increased to 39% with ARA-1 and 48% with ARA-2. Other authors reported higher xylose yields, but with the participation of additional accessory hemicellulases as α-glucuronidases [16]. While ARA-1 has higher efficiency in degrading arabinoxylan compared to ARA-2, the results indicate that ARA-2 addition (Figure 7, at the bottom) releases a greater amount of xylose when used as a debranching enzyme. This could be attributed to the lower stability of ARA-1, which causes it to rapidly lose activity at 50 °C. As a result, ARA-2, which is much more robust, can continue degrading arabinoxylan for longer, and the combination of enzymes ultimately produces more xylose.

## 3. Materials and Methods

### 3.1. Nucleic Acid Isolation and Cloning of Two α-l-Arabinofuranosidases

The α-l-arabinofuranosidases genes coding for ARA-1 (XM_040874772.1) and ARA-2 (XM_040875155.1) enzymes were obtained from the previously sequenced and annotated *T. amestolkiae* genome [8]. Based on these sequences, primers were designed excluding the signal peptide (using signalP 5.0 server [40]) and introducing the necessary cutting sites for the restriction enzymes EcoRI and NotI (ARA-1Fw 5′-CTAGAATTCCTTCCTACGGATGAAACTGCAC-3′, ARA-1Rv 5′-ATTGCGGCCGCCTATTTGTGGGTAAGTAAACCTGGA-3′, ARA-2Fw 5′-CAAGAATTCCAAGCATCGCTTTATGGTCAATG-3′, ARA-2Rv 5-ATTGCGGCCGCCTACTTGGACAGGGTCAAGAG-3′). RNA extraction of the fungus and conversion into cDNA was performed, as explained in previous works [12], in order to check the presence of introns. Genes were amplified by PCR using cDNA as a template and cloned into the pPIC9 vector. Plasmids were propagated in *Escherichia coli* DH5α and isolated with the High Pure Plasmid Isolation Kit (Roche, Basel, Switzerland). After confirming the correct sequence, the vector was linearized with SacI (New England Biolabs, Ipswich, MA, USA) and used to transform *P. pastoris* GS115, following the protocol explained in previous works [41].

### 3.2. Production and Purification of ARA-1 and ARA-2

The preinocula of the selected clones were created in 50 mL Falcon tubes with 5 mL of YEPS medium (10 g/L yeast extract, 20 g/L peptone, 10 g/L sorbitol, 100 mM potassium phosphate buffer, pH 6) and used to inoculate 250 mL flasks with 25 mL of YEPS medium. These cultures were incubated at 28 °C and 250 rpm adding a daily dose of methanol (at a final concentration in the culture medium of 0.5% *w*/*v*) to induce enzyme expression. Every day, 1 mL of sample was taken from each culture to analyze growth (optical density at 600 nm) and enzyme activity. Once the most productive clone was selected, new cultures were carried out following the previous protocol, but scaling them to 200 mL of YEPS medium in 1 L flasks. All the measurements were performed in triplicate. Culture supernatants containing enzymatic crudes were collected when maximal activity was obtained (Figure 1, day 8 for ARA-1 and day 9 for ARA-2).

The standard reaction for detection of α-l-arabinofuranosidase activity was determined spectrophotometrically by measuring the release of *p*-nitrophenol (*p*NP) (ε410 = 15,200 M^−1^ cm^−1^) from the substrate *p*-nitrophenyl-α-l-arabinofuranoside (Sigma-Aldrich, St. Louis, MO, USA). A final volume of 200 μL was used, containing the appropriate amount of crude or pure enzyme, bovine serum albumin (BSA) at a final concentration of 0.1%, 3.5 mM *p*-nitrophenyl-α-l-arabinofuranoside and 50 mM sodium acetate buffer at pH 5. The reaction was incubated for 10 min at 50 °C and 1200 rpm and stopped with 500 μL of 2% (*w*/*v*) sodium carbonate. The release of *p*NP was measured at 410 nm. One unit of activity was defined as the amount of enzyme capable of releasing 1 μmol of *p*NP per min. BSA was added to every reaction in order to obtain reproducible results and to prevent loss of enzyme activity [42].

For protein purification, both *P. pastoris* culture supernatants containing ARA-1 and ARA-2, respectively, were centrifuged at 5000× *g* and 4 °C for 20 min. The supernatants were vacuum filtered through 0.8, 0.45, and 0.22 µm membranes (Merck-Millipore, Burlington, VT, USA) and dialyzed and concentrated by tangential filtration and ultrafiltration using 10 kDa membranes. The enzymes were purified from their respective crude enzyme extracts by fast protein liquid chromatography (FPLC) in an ÄKTA Purifier chromatographic system (GE Healthcare, Chicago, IL, USA) using the same protocol. The dialyzed crudes were loaded in a Hi-Trap QFF anion exchange cartridge (GE Healthcare) and equilibrated in 10 mM sodium phosphate buffer (pH 6). After elution of not retained proteins (including α-l-arabinofuranosidase activity), the column was washed with 1 M NaCl (removing bound proteins) and re-equilibrated with the starting buffer. The flow rate was 2 mL/min. Fractions with α-l-arabinofuranosidase activity were collected, dialyzed, and concentrated by ultrafiltration using Amicon Ultra-15 units (Merck-Millipore). The homogeneity of the purified proteins, and their molecular masses, were analyzed in polyacrylamide gels with sodium dodecyl sulfate (SDS-PAGE), as described below. The purified enzymes were stored at 4 °C, maintaining the activity for at least 6 months. Protein concentration was determined using a Nanodrop spectrophotometer with the default settings (1 Abs = 1 mg/mL) after checking that similar values were obtained by BCA assay using the BCA protein assay kit (Thermo Scientific, Waltham, MA, USA) with bovine serum albumin as standard.

### 3.3. Determination of the Physicochemical Properties of the Recombinant ARA-1 and ARA-2

A total of 10 µg of each protein was loaded on a 10% SDS-PAGE gel and, after electrophoresis, stained with Coomassie blue R-250 (Sigma-Aldrich). In addition, samples treated with endoglycosidase H (Endo H, Roche), following the manufacturer’s instruction, were also analyzed to determine the amount of N-linked carbohydrates from the difference of the proteins’ molecular mass before and after deglycosylation. The accurate molecular masses of the pure enzymes were assessed by MALDI-TOF following the protocol of Mendez-Líter et al. [41].

The influence of temperature and pH on the stability and optimal reaction conditions of both purified enzymes was studied. pH was analyzed in a range from 2.2 to 7 for optimal activity and from 2.2 to 9 for stability. The buffers selected for each pH segment were glycine-HCl (pH 2.2–3), sodium formate (pH 3–4), sodium acetate (pH 4–5.5), histidine-HCl (pH 5.5–7), and Tris-HCl (pH 8–9), all of them at 50 mM, with 0.1% BSA to prevent activity loss. The reaction parameters set for determination of the optimum pH were the same as those used in the standard reaction except for the buffer, while for pH stability, the samples were incubated at 4 °C for 72 h. The optimum temperature was determined in 10 min reactions under standard conditions, but changing the reaction temperature from 30 to 70 °C. The thermostability of the enzyme was analyzed by incubating at 30, 40, 50, 60, and 70 °C. The activity was periodically determined as described above. The highest activity values determined in the assays of optimal temperature or pH were taken as 100%.

The effect of common chemical compounds on the arabinofuranosidases was also analyzed: LiCl, KCl, AgNO_3_, MgSO_4_, CaCl_2_, BaCl_2_, MnCl_2_, FeSO_4_, CoCl_2_, NiSO_4_, CuSO_4_, ZnSO_4_, HgCl_2_, Pb(NO_3_)_2_, AlNH_4_(SO_4_)_2_, NH_4_(SO_4_)_2,_ FeSO_4,_ and ethylenediaminetetraacetic acid (EDTA) were added to the reaction mixture a final concentration of 5 mM, and 2-mercaptoethanol (2-ME) and dithiothreitol (DTT) at a concentration of 10 mM. The assay was carried out under standard conditions, and 100% activity corresponds to reactions without chemicals addition. 

Finally, the cellulose-binding capacity of each α-arabinofuranosidase was determined since the presence of a cellulose-binding domain was detected in the ARA-2 sequence. For this, the enzymes were incubated in a 1% (*w*/*v*) solution of crystalline cellulose at 4 °C, with 0.1% BSA, in 50 mM pH 5 sodium acetate buffer and 1200 rpm for 2 h. Subsequently, samples were taken, centrifuged to precipitate the polysaccharide, and the activity in the supernatant was measured to determine the amount of unbound enzyme. Assays were carried out under standard conditions, and 100% activity was determined by incubating the enzyme without polysaccharide. 

### 3.4. Substrate Specificity of Recombinant ARA-1 and ARA-2

The kinetic parameters for each substrate were calculated by measuring the enzymatic activity against *p*-nitrophenyl-α-l-arabinofuranoside (0.0625 to 16 mM), wheat arabinoxylan (0.023% to 3% *w*/*v*) and arabinan (0.0625–16% *w*/*v*). In the case of the last two polysaccharides (obtained from Megazyme, Bray, Ireland), the activity was assessed by quantifying the release of L-arabinose, using the L-arabinose/D-galactose assay kit (Megazyme), following the manufacturer’s instructions. The activity data were fitted to the Michaelis–Menten kinetic model with the SigmaPlot version 12.3 program (Systat Software). One unit of activity was defined as the amount of enzyme capable of releasing 1 μmol of arabinose per min.

### 3.5. XOS Release from Arabinoxylan and Saccharification

The effect of *T. amestolkiae* arabinofuranosidases on XOS release from wheat arabinoxylan was studied by means of an enzymatic cascade using the endoxylanase XynM and β-xylosidase BxTw1 from *T. amestolkiae*, heterologously produced in *P. pastoris* and purified as previously described [10,43]. Wheat arabinoxylan, from Megazyme (0.7 % *w*/*v* final concentration), was preincubated with 0.75 μg of ARA-1 or ARA-2 in 1 mL sodium acetate pH 4, 50 mM, including 0.05% BSA at 50 °C and 1200 rpm for 10 min. After preincubation, 0.45 μg/mL of the recombinant XynM endoxylanase from *T. amestolkiae* was added, and the reaction was incubated at 50 °C and 1200 rpm for 18 h.

The effect of arabinofuranosidases on arabinoxylan saccharification was also studied in the same conditions but with the simultaneous addition of XynM endoxylanase (0.45 μg/mL) and the recombinant BxTw1 β-xylosidase from *T. amestolkiae*. 0.86 μg/mL of BxTw1 was initially added, and after 18 h of reaction, an excess of BxTw1 (5× = 4.3 μg/mL) was added to increase the saccharification of arabinoxylan into xylose and incubated for 96 h. 

In both cases, for identification of the reaction products, samples were centrifuged at 14,000× *g*, filtered through 0.22 μm filters, and injected into an Aminex HPX-42C column using an Agilent 1260 series equipped with a refraction index detector (RID) for xylooligosaccharides and arabinose detection. Distilled water was used as eluent for isocratic chromatography at 0.5 mL/min, with column temperature set at 85 °C and RID temperature at 55 °C. Standard solutions of xylotriose, xylotetraose, and xylopentaose at 5 μg/mL–1 mg/mL (Megazyme) or arabinose at 50 μg/mL–50 mg/mL (Merck, Rahway, NJ, USA) were prepared using distilled water. Xylose and xylobiose quantification were also performed by GC/MS. To do so, samples were mixed with 100 µg/mL of myo-inositol as an internal standard, dried, and transformed into trimethylsilyl oximes. The dry sample was treated with 250 µL of a solution of hydroxylamine chloride in pyridine (70 °C, 30 min), and then 150 µL of BSTFA was added, maintaining the reaction at 80 °C for 10 min. Identification and quantification were performed on an Agilent 6890A/5975 GC/MS, using a DB-5HT column (30 m × 0.25 mm I.D. × 0.1 µm film thickness) with He as carrier gas. The samples were analyzed with a temperature program: 160 °C for 2 min, then 2 °C min^−1^ up to 185 °C. The identification was achieved by comparing the retention time and mass spectrum of the sample components with those previously determined for standards analyzed under identical conditions.

## 4. Conclusions

In this work, two novel α-l-arabinofuranosidases have been heterologously produced and characterized. The catalytic efficiency of ARA-1 toward arabinoxylan is one of the highest found in the literature, although its pH and temperature tolerances are lower than those described for ARA-2. Both α-l-arabinofuranosidase acted synergistically with an endoxylanase from the same fungus the highest degree of synergy found between those described to date) and have been successfully applied to convert arabinoxylan into xylooligosaccharides. Depolymerization of the polysaccharide to fermentable sugars was accomplished by using an enzymatic cascade combining recombinant α-l-arabinofuranosidase, endoxylanase, and β-xylosidase from *T. amestolkiae.*

## Figures and Tables

**Figure 1 ijms-24-11997-f001:**
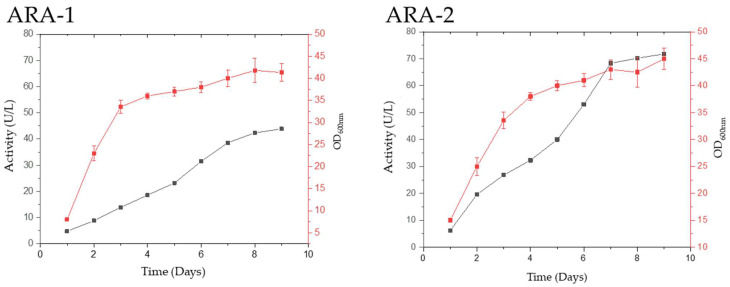
*P. pastoris* recombinant strains producing α-l-arabinofuranosidases. Red line, optical density at 600 nm; black lines, enzymatic activity against *p*-nitrophenyl α-l-arabinofuranoside.

**Figure 2 ijms-24-11997-f002:**
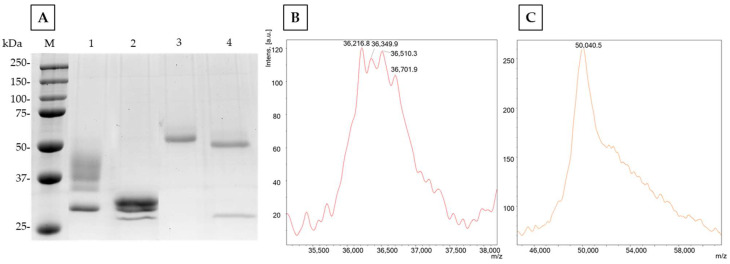
Estimation of the molecular mass of ARA-1 and ARA-2. (**A**) SDS-PAGE: M, molecular mass marker; lanes 1 and 3, ARA-1 and ARA-2, respectively; lanes 2 and 4, ARA-1 and ARA-2 deglycosylated with EndoH, respectively (Endo H band appeared around 27 kDa). (**B**,**C**) MALDI-TOF analysis of ARA-1 and ARA-2, respectively.

**Figure 3 ijms-24-11997-f003:**
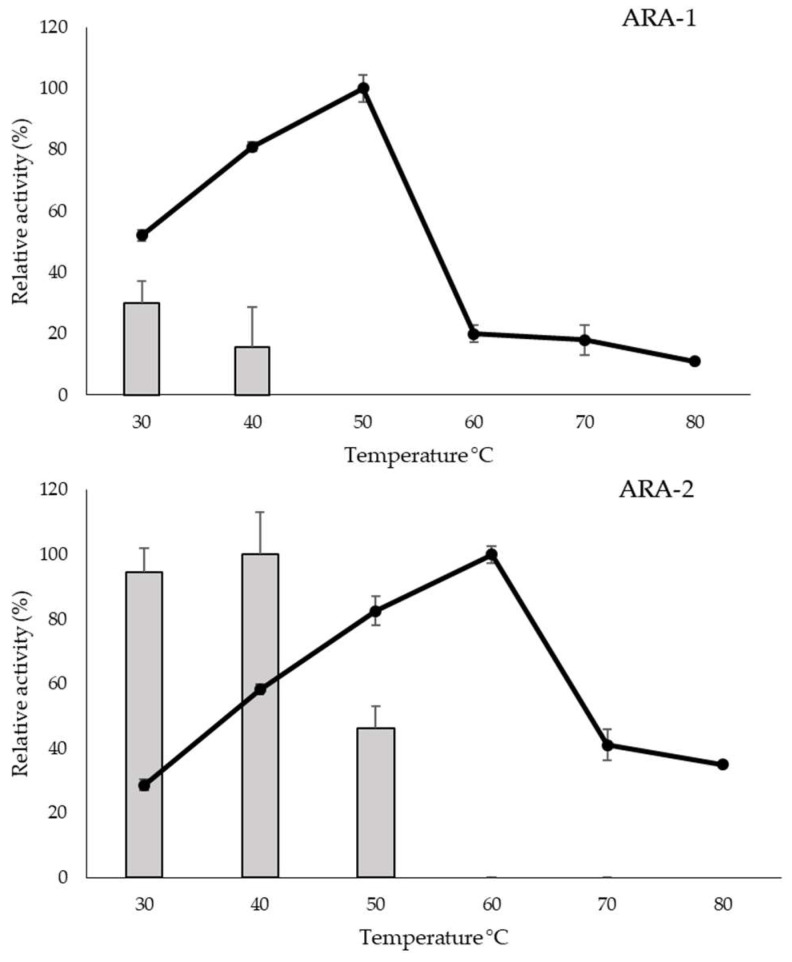
Effect of temperature on *T. amestolkiae* arabinofuranosidases activity. The lines indicate the optimum temperature for enzyme activity; bars show 72 h stability in a range of temperatures from 30 °C to 80 °C. A total of 100% corresponded to the highest activity value observed in this assay.

**Figure 4 ijms-24-11997-f004:**
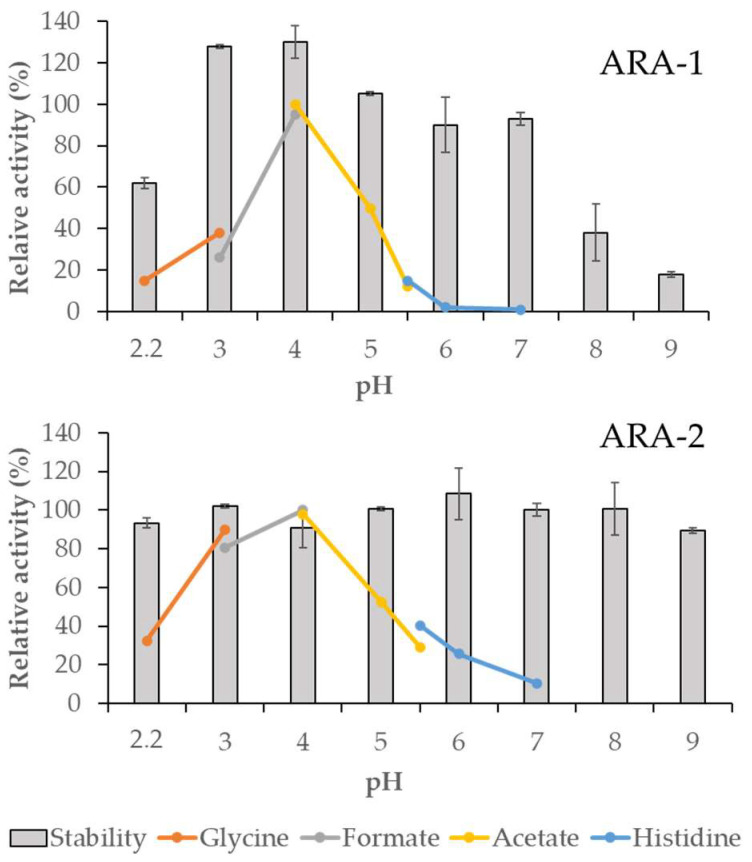
Effect of pH on the activity of *T. amestolkiae* arabinofuranosidases. The lines indicate the optimum pH for enzyme activity in different buffers; bars show 72 h stability in a pH range from 2 to 9. 100% corresponded to the highest activity value observed in this assay. Glycine: glycine-HCl buffer; formate: sodium formate buffer; acetate: sodium acetate buffer; histidine: histidine-HCl buffer.

**Figure 5 ijms-24-11997-f005:**
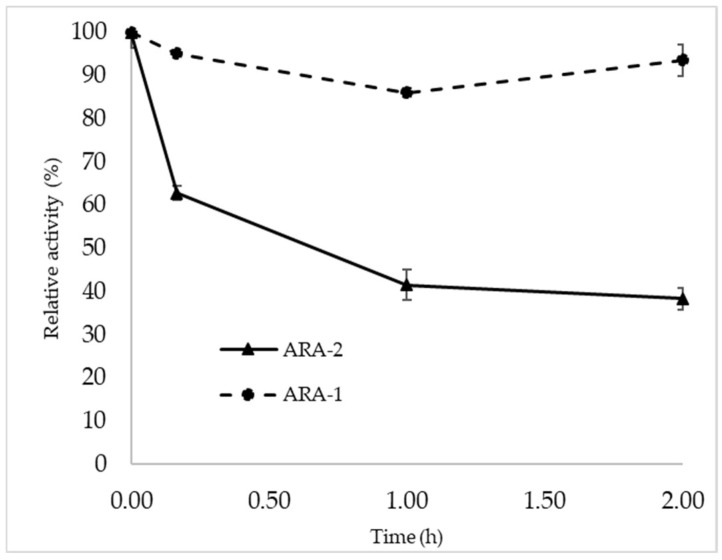
Cellulose-binding assays. The arabinofuranosidases were incubated with crystalline cellulose for 2 h at 4 °C, and residual activity was measured in the supernatants. In these assays, the activity was normalized with respect to the value observed for each enzyme in reactions without cellulose, which was considered 100%.

**Figure 6 ijms-24-11997-f006:**
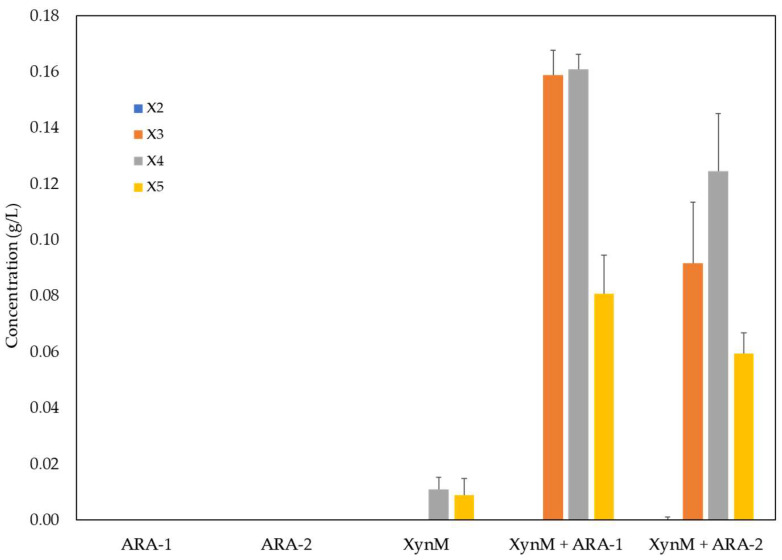
Xylooligosaccharides and xylose liberation from arabinoxylan. The reaction was catalyzed by endoxylanase XynM from *T. amestolkiae* (18 h of incubation) after preincubation with ARA-1 or ARA-2 (10 min). Control reactions were performed without ARA-1 or ARA-2 pretreatment. X2, xylobiose; X3, xylotriose; X4, xylotetraose; X5, xylopentaose.

**Figure 7 ijms-24-11997-f007:**
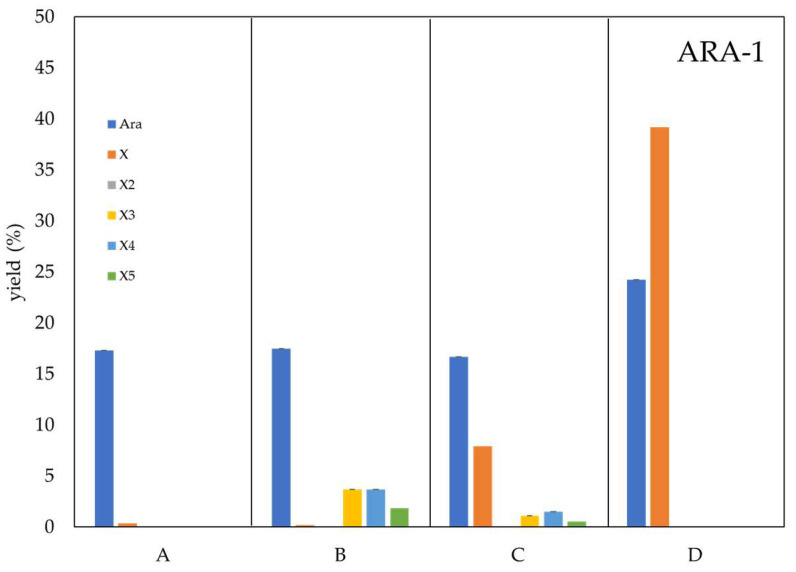
Effect of ARA-1 and ARA-2 pretreatment on arabinoxylan saccharification. Samples were pretreated with ARA-1 or ARA-2 (10 min) and then incubated for 18 h with: (A) BxTw1 (0.86 μg/mL); (B) XynM (0.45 μg/mL); (C) XynM and BxTw1; (D) the samples from C were supplemented with an excess of BxTw1 (5×, 96 h) to help increasing xylose liberation. X, xylose; X2, xylobiose; X3, xylotriose; X4, xylotetraose; X5, xylopentaose.

**Table 1 ijms-24-11997-t001:** Recovered activity and degree of purification of ARA-1 and ARA-2.

ARA-1			
Step	Specific Activity (U/g)	Yield (%)	Purification Fold
Crude extracts	16.9	100	-
Purified enzyme	67.5	73	4
**ARA-2**			
**Step**	**Specific Activity (U/g)**	**Yield (%)**	**Purification Fold**
Crude extracts	200	100	-
Purified enzyme	890	71.8	4.5

**Table 2 ijms-24-11997-t002:** Physico-chemical properties and kinetic properties of several GH62 arabinofuranosidases.

Organism	Enzyme	Optimum pH	pH Stability	Optimum Temperature	Temperature Stability	Wheat Arabinoxylan	*p*NP-AF	Arabinan
						*k_cat_/K_m_* (s^−1^ mg^−1^ mL)	Specific Activity (U/mg)	Specific Activity (U/mg)
*Eupenicillium parvum* [16]	EpABF62A	4.5	2–11	55	50 (24 h)	12	0.4	-
*Aspergillus nidulans* [17]	AnAbf62A-m2,3	5.5	-	70	-	36.3	1.66	1.43
*Streptomyces* sp. SWU10 [18]	WUAbf62A	5	4–9	50	50 (1 h)	0.15 *	-	0.01 *
*Nectria haematococca* [19]	NhaAbf62A	6.5	-	40	40 (45 min)	6.1	-	0.4
*Sporisorium reilianum* [19]	SreAbf62A	6.5	-	40	50 (45 min)	3.3	-	0
*Gibberella zeae* [19]	GzeAbf62A	6.5	-	40	40 (45 min)	8.2	-	0
*Penicillium funiculosum* [20]	ABF62a	3.5	-	40	-	7.25	-	-
	ABF62b	2.2	-	40	-	153.6	-	-
	ABF62c	3	-	50	-	55.1	-	-
*Penicillium subrubescens* [21]	AxhA	5	4–8	40	60 (1 h)	-	4.1	-
	AxhB	5	3–8	40	50 (1 h)	-	14.6	-
	AxhC	5	3–7	40	50 (1 h)	-	3.8	-
	AxhD	4	2–7	40	50 (1 h)	-	10.9	-
*Scytalidium thermophilum* [22]	Abf62A	4–5	-	50	-	5.5	0.02	
	Abf62C	6–7	-	50	-	4.5	0.24	
*Talaromyces amestolkiae* (this work)	ARA-1	4	3–7	50	40 (72 h)	129	0.07	0.3
	ARA-2	4	2–9	60	50 (72 h)	9.60	0.5	0.28

* Calculated from data from the original article.

**Table 3 ijms-24-11997-t003:** Kinetic parameters of ARA-1 and ARA-2 against different substrates.

Substrate		ARA-1	ARA-2
***p*NP-AF**	*K_m_* (mM)	8.97 ± 0.06	1.48 ± 0.08
	*k_cat_* (s^−1^)	0.05 ± 0.001	0.41 ± 0.01
	*k_cat_*/*K_m_* (mM^−1^·s^−1^)	0.005 ± 0.0004	0.28 ± 0.01
**Arabinoxylan**	*K_m_* (mg/mL)	0.29 ± 0.11	1.80 ± 0.43
	*k_cat_* (s^−1^)	37.17 ± 1.97	16.89 ± 1.23
	*k_cat_*/*K_m_* (mg^−1^·mL·s^−1^)	129.2 ± 3.2	9.60 ± 0.27
**Arabinan**	*K_m_* (mg/mL)	1.82 ± 0.23	3.4 ± 0.42
	*k_cat_* (s^−1^)	0.20 ± 0.01	0.23 ± 0.01
	*k_cat_*/*K_m_* (mg^−1^·mL·s^−1^)	0.11 ± 0.001	0.07 ± 0.01

## Data Availability

Data are contained within the article and the Appendix A.

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
