# Peer review of "Expression and Characterization of Two α-l-Arabinofuranosidases from Talaromyces amestolkiae: Role of These Enzymes in Biomass Valorization"

_ijms, 2023, doi:10.3390/ijms241511997_

Round 1

Reviewer 1 Report

Dear Authors, 

Thank you for submitting your manuscript to IJMS. The manuscript does presents a great study on the thermally stable arabinofuranosidases. However, at several instances the information is missing or not well presented. Please use standard units throughout the manuscript. At several instances the units used are not standard viz. mU/mL. Please see below:
1. Table 2: Why are the enzymatic units not available for the other enzymes mentioned in table. Merely comparing the physio-chemical parameters cannot substantiate the comparison. Also, no units for temperature
2. Why the enzymatic activity results are presented in mU/mL. The enzyme activity should be reported in standard units. (Units/unit volume)
3. Why is the temperature stability reported in 72 hours? Why the half life for the enzymes not reported that will provide the data for the all the temperatures tested? 
4. No mention of the 100% activity in the graphs or texts for each parameters tested? How to determine if the enzymatic activity increased or decreased and by how much extent with each variable tested?
5. How long the production of enzymes ran? There is a mention of addition methanol and removing samples daily but no mention of for how many days? 
6. Line 353: The activity was maintained for 6 months. More elaboration is required?
can refer to https://www.mdpi.com/2076-2607/6/3/93 for some details. 

Thanks

Author Response

Point by point responses to reviewers.

Reviewer 1:

Dear Authors, 

Thank you for submitting your manuscript to IJMS. The manuscript does present a great study on the thermally stable arabinofuranosidases. However, at several instances the information is missing or not well presented. Please use standard units throughout the manuscript. At several instances the units used are not standard viz. mU/mL. Please see below:

Thank you for the useful comments that will help in the rebuilding of our manuscript. The international unit is defined in M&M.

  1. Table 2: Why are the enzymatic units not available for the other enzymes mentioned in table. Merely comparing the physio-chemical parameters cannot substantiate the comparison. Also, no units for temperature.

Table 2 main objective was gathering the information about the optimal pH and temperature values of all fungal enzymes from GH62 family. However, usually this information is included in the articles as a relative value, with respect to the total activity in optimal conditions. Thus, it cannot be compared between manuscripts since they are not absolute values. Nevertheless, in table XX we had included the catalytic values of these enzymes versus different substrates. In the new version of the manuscript, we have merged both tables, including both information about optimal conditions and catalytic parameters, as suggested by reviewer 1.

  1. Why the enzymatic activity results are presented in mU/mL. The enzyme activity should be reported in standard units. (Units/unit volume).

Enzymatic activity has been changed to U/L as suggested. The definition of activity unit is included in M&M.

  1. Why is the temperature stability reported in 72 hours? Why the half life for the enzymes not reported that will provide the data for the all the temperatures tested? 

We decided to study the temperature stability after 72 h of incubation because, like many other industrial applications, our process (XOS release and saccharification) takes several days. We considered that using conditions similar to those applied in the process is the most convenient way to test the stability of the enzymes.

  1. No mention of the 100% activity in the graphs or texts for each parameters tested? How to determine if the enzymatic activity increased or decreased and by how much extent with each variable tested?

We apologize for the mistake. We have added the description of 100% in all graphs where needed.

  1. How long the production of enzymes ran? There is a mention of addition methanol and removing samples daily but no mention of for how many days? 

The original version of the manuscript referred to these points in the Results and Discussion section (line 108-109 Enzymatic crudes, containing ARA-1 or ARA-2, were obtained from P. pastoris culture supernatants when it showed maximal activity and loaded into the anion exchange cartridge (Figure S2)). However, we have also included an explanatory statement in the methods as suggested by the reviewer.

  1. Line 353: The activity was maintained for 6 months. More elaboration is required?
    can refer to https://www.mdpi.com/2076-2607/6/3/93 for some details. 

Although we did not perform a detailed long-term experiment to analyze the stability of enzymes, we periodically controlled their activity. Hence, it can be stated that activity was maintained at least for 6 months.

Reviewer 2 Report

It is always nice to see new work on biochemical characterisation of enzymes. Regrettable, the scope of this work is very limited. What was the hypothesis behind the work? What is the main new finding?

When 24 similar enzymes has already been characterised, it is very difficult to see what we learn from number 25 and 26. It is misleading to state in the abstacr that a process has been designed, when you have simply tested the effect of using a mixture of enzymes that are already known. This is a study of substrate specificity.

The quality of the language is good. However, there is a strange choice of word many places such as l13 "rupture", l29 "eliminate", L43 "pool"l252 "ramifications".

More attention is also need to the figure legends e.g. fig 3 what is 100% and fig 4 what is Glicine?

Author Response

Point by point responses to reviewers.

Reviewer 2:

Comments and Suggestions for Authors

1. It is always nice to see new work on biochemical characterisation of enzymes. Regrettable, the scope of this work is very limited. What was the hypothesis behind the work? What is the main new finding?

When 24 similar enzymes has already been characterised, it is very difficult to see what we learn from number 25 and 26. It is misleading to state in the abstacr that a process has been designed, when you have simply tested the effect of using a mixture of enzymes that are already known. This is a study of substrate specificity.

We thank reviewer 2 for his/her comments and agree that when describing a new enzyme belonging to a well-characterized family it is necessary to describe some novel feature. In fact, the search for new enzymes continues in order to find catalysts with better properties to apply them in new procedures or to improve existing processes. In this sense, ARA-1 is one of the most active arabinofuranosidases described to date, only surpassed by ABF62b from P. funiculosum (Table 2). Likewise, when analyzing the effect of the addition of ARA from T. amestolkiae on the release of XOS, the degree of synergy found is much higher than those described in the literature. This feature has been highlighted in the new version of the manuscript, at the end od the summary and the conclusions.

Comments on the Quality of English Language

1.- The quality of the language is good. However, there is a strange choice of word many places such as l13 "rupture", l29 "eliminate", L43 "pool"l252 "ramifications".

“Rupture” was replaced by “break”, “eliminate” was deleted due to rephrasing, “pool” was replaced by “variety”, “ramifications” was replaced by “substitutions”.

2. More attention is also need to the figure legends e.g. fig 3 what is 100% and fig 4 what is Glicine?

Definition of 100% value has been added to the figure legend and text as suggested by reviewers. An explanatory sentence has been added in fig 3 legend. “Glicine” is Glycine, it was a writing mistake.